# Ambiguity about Splicing Factor 3b Subunit 3 (SF3B3) and Sin3A Associated Protein 130 (SAP130)

**DOI:** 10.3390/cells10030590

**Published:** 2021-03-08

**Authors:** Paula I. Metselaar, Celine Hos, Olaf Welting, Jos A. Bosch, Aletta D. Kraneveld, Wouter J. de Jonge, Anje A. Te Velde

**Affiliations:** 1Tytgat Institute for Liver and Intestinal Research, AGEM, Amsterdam UMC, University of Amsterdam, 1105BK Amsterdam, The Netherlands; c.hos@student.vu.nl (C.H.); o.welting@amsterdamumc.nl (O.W.); w.j.dejonge@amsterdamumc.nl (W.J.d.J.); a.a.tevelde@amc.nl (A.A.T.V.); 2Department of Psychology, University of Amsterdam, 1018WS Amsterdam, The Netherlands; j.a.bosch@uva.nl; 3Department of Medical Psychology, Amsterdam UMC, University of Amsterdam, 1001NK Amsterdam, The Netherlands; 4Division of Pharmacology, Utrecht Institute for Pharmaceutical Sciences, Faculty of Science, Utrecht University, 3584CG Utrecht, The Netherlands; a.d.kraneveld@uu.nl

**Keywords:** SF3B3, SAP130, splicing factor 3b subunit 3, Sin3A associated protein 130, spliceosome-associated protein 130

## Abstract

In 2020, three articles were published on a protein that can activate the immune system by binding to macrophage-inducible C-type lectin receptor (Mincle). In the articles, the protein was referred to as ‘SAP130, a subunit of the histone deacetylase complex.’ However, the Mincle ligand the authors aimed to investigate is splicing factor 3b subunit 3 (SF3B3). This splicing factor is unrelated to SAP130 (Sin3A associated protein 130, a subunit of the histone deacetylase-dependent Sin3A corepressor complex). The conclusions in the three articles were formulated for SF3B3, while the researchers used qPCR primers and antibodies against SAP130. We retraced the origins of the ambiguity about the two proteins and found that Online Mendelian Inheritance in Man (OMIM) added a Nature publication on SF3B3 as a reference for Sin3A associated protein 130 in 2016. Subsequently, companies such as Abcam referred to OMIM and the Nature article in their products for both SF3B3 and SAP130. In turn, the mistake by OMIM followed in the persistent and confusing use of ‘SAP130′ (spliceosome-associated protein 130) as an alternative symbol for SF3B3. With this report, we aim to eliminate the persistent confusion and separate the literature regarding the two proteins.

## 1. Introduction

There is persistent confusion regarding splicing factor 3b subunit 3 (SF3B3), which has been investigated for its function as a ligand of the macrophage-inducible C-type lectin receptor (Mincle). In the literature, the protein is referred to by approved symbol SF3B3, or by the alternative name of spliceosome-associated protein 130 (abbreviated as SAP130). However, SAP130 is also the HUGO Gene Nomenclature Committee (HGNC) approved symbol of a different protein, Sin3A associated protein 130 (SAP130), a Sin3A corepressor complex component unrelated to SF3B3. In 2020, three articles were published [1,2,3] on Mincle ligand SF3B3, in which the protein was referred to as ‘SAP130, a subunit of the histone deacetylase complex. Moreover, the research materials used were designed and selected for SAP130, not for SF3B3. It is clear that the authors were not aware of the difference between the two proteins. We could retrace this confusion to the use of the alternative symbol ‘SAP130′ for SF3B3 since 2000 and the approved symbol SAP130 for Sin3A associated protein 130 since 2006. The confusion became a mistake in 2016, when Online Mendelian Inheritance in Man (OMIM) [4], a much frequented database by researchers, referred to an article published in Nature [5] on Mincle ligand SF3B3 in their entry on SAP130. This article [5] described SF3B3 as ‘SAP130′, a subunit of the histone deacetylase complex.’ Here, we performed a literature review of both proteins and carefully checked the use of gene and protein names, abbreviations, references, and methods sections to determine the extent of the problem. We conclude that SF3B3 and SAP130 are two distinct proteins and should be referred to in the existing and future literature according to the official nomenclature only.

## 2. Materials and Methods

*Literature review.* We performed a literature review in PubMed to retrieve all articles describing SAP130 or SF3B3 with the query: (SF3B3[All Fields] OR SAP130[All Fields] OR “spliceosome associated protein 130”[All Fields] OR “splicing factor 3b subunit 3”[All Fields] OR Sin3A associated protein 130), on 29 April 2020. We last updated the query results on 20 January 2021. Relevant articles we came across during the review were included. We selected both articles and reviews that were written in English. For each paper, we carefully checked the use of the gene or protein name and abbreviation, references to earlier research, and the materials and methods section (e.g., primer sequences and antibody ascension numbers).

*Contact with selected corresponding authors.* If the materials and methods could not be checked due to missing information, in addition to the use of an incorrect abbreviation, we sent a standardized email to the corresponding author to request the missing information such as primer sequences and/or antibody numbers. After two weeks, a reminder was sent to the six authors who did not respond. Furthermore, the corresponding authors of papers in which mistakes were found were notified via email.

*Immunohistochemistry.* Paraffin-embedded tissues were sectioned at 4 µm and mounted on slides. Samples were de-paraffinized in a xylene and ethanol sequence and incubated for 20 min in a solution of hydrogen peroxide methanol (0.1% H_2_O_2_ in MeOH) to reduce endogenous peroxidases. Antigen retrieval was performed in Tris/EDTA buffer (pH = 9.0) at 120 °C for ten minutes. Subsequently, samples were blocked by PBT for 30 min and incubated with the primary antibody overnight at 4 °C (anti-SF3B3 antibody: Abcam, Ab209402, LOTGR2554557-3, diluted at 1:32,000 in PBT; anti-SAP130 antibody: Abcam, Ab111739, LOTGR168825-6, diluted at 1:100 in PBT). Next, samples were incubated with secondary antibody (BrightVision from Immunologic, Poly-HRP-anti rabbit IgG, LOT210219, undiluted) for 30 min. Afterwards, samples were incubated with DAB substrate for two minutes, counterstained with hematoxylin for four minutes, then dehydrated, cleared and mounted. Two negative controls, without primary or secondary antibody, were treated equally. Imaging was performed on the Olympus BX-51 widefield microscope at 20× and 40× magnification.

*Ethical considerations.* Inflamed human ileum epithelium of a patient with Crohn’s Disease was acquired from the surgical department with signed informed consent prior to surgery, as reviewed and approved by the “Medisch Ethische Toetsings Commissie” (METC) AMC Amsterdam (institutional ethics committee at the Academic Medical Center Amsterdam; reference: METC #2014_178). Healthy human liver tissue was collected anonymously by the pathology department of the AMC Amsterdam according to the code of conduct for responsible use, under a waiver granted by the METC (reference: W12_216 # 12.17.0246).

## 3. Results

SF3B3 was first described by Das et al. in 1999 [6] as a component of U2 small nuclear ribonucleoprotein-associated protein complex SF3b. The authors isolated cDNA that encoded a 130kDa protein of 1217 amino acids and named SF3B3 “spliceosome-associated protein 130” (SAP130) because it is part of a spliceosome complex. The protein is primarily found in the nucleus. Its coding gene is located on chromosome 16. Among the most notable discoveries is the fact that SF3B3 is part of DNA repair and RNA transcription complexes. SF3B3 is similar in sequence and structure to the large subunit of UV-damaged DNA-binding protein (UV-DDB p127), which is implicated in the repair of DNA lesions. SF3B3 is also a subunit of TBP-free TAF(II) complex (TFTC), a transcription initiator. SF3B3, and therefore TFTC, can bind UV-damaged DNA [7]. SPT3-TAF(II)31-GCN5L acetylase (STAGA, a transcription coactivator-histone acetyltransferase complex) associates with SF3B3 and UV-DDB p127, suggesting functions of STAGA in transcription and DNA repair [8]. Finally, SF3B3 forms complexes with active cullin-RING E3 ligases, which are critical for ubiquination [9]. Then, in 2008, Yamasaki et al. [10] found that SF3B3 could activate macrophage-inducible C-type lectin receptor (Mincle). To date, SF3B3 is most frequently studied regarding its function as a danger-associated molecular pattern (DAMP) that can activate the immune system by binding to Mincle. In all articles mentioned above, SF3B3 was referred to as spliceosome-associated protein 130 and its unfortunate abbreviation SAP130, even though SF3B3 was already approved as the gene symbol by HGNC in February 2000.

In 2003, Fleischer et al. [11] identified a subunit of the histone deacetylase-dependent Sin3A corepressor complex which consists of 1048 amino acids and is 130kDa and therefore is called Sin3A associated protein 130 (SAP130). Its encoding gene is located on chromosome 2. HGNC approved SAP130 as the official gene symbol of Sin3A associated protein 130 in February 2006. Using antibodies directed at SF3B3 and SAP130, we can see distinguishable staining patterns in two different tissues (Figure 1). With no background staining in negative controls (not shown here) and at similar staining intensity, SF3B3 is visible only in the nucleus, while SAP130 seems to be present more ubiquitously.

Regretfully, this distinction was missed by three papers published in 2020. Two research articles, published by the same research group [1,2], used the abbreviation SAP130 for SF3B3 and designed qPCR primers targeted at Sin3A associated protein 130 rather than SF3B3. The third research article [3] also referred to SF3B3 as ‘Spliceosome-associated protein 130 (SAP130), a subunit of histone deacetylase’ and used an antibody against Sin3A associated protein 130 instead of SF3B3. The authors of the papers published in Annals of the New York Academy of Sciences [1], Journal of Crohn’s & Colitis [2] and Annals of Translational Medicine [3] apparently were not aware of the distinction between the two proteins, and ordered and designed materials by looking for SAP130, which returns both proteins. We have notified the authors about the confusion of SF3B3 and SAP130 but received no response.

We have traced the confusion of the two proteins back to its origin. Since the description of SF3B3 as SAP130 in 1999, and the discovery of SAP130 in 2003, the same abbreviation has been used for both proteins. It was a matter of time until researchers confused the information of one protein with the other. In November 2013, two papers were published, a research article [12] in Scientific Reports and a review [13] in International Immunopharmacology that investigated SF3B3. The research article did not use the full name of the protein, only the abbreviation SAP130, and mistakenly described it as a subunit of histone deacetylase. The review incorrectly referred to SF3B3 as Sin3A associated protein 130 (SAP130). In 2016, a research article was published in Nature [5] that also described SF3B3 as ‘SAP130, a subunit of the histone deacetylase complex.’ OMIM [4] erroneously cited this SF3B3-article in its page about Sin3A associated protein 130. In turn, OMIM is used as a reference database for reagent suppliers such as Abcam, Abnova, MyBioSource and Santa Cruz Biotechnologies. In their product descriptions, SAP130 and SF3B3 are used interchangeably, summarizing information on both proteins and even including both accession numbers. Consequently, researchers may unknowingly purchase the incorrect materials, and study Sin3A corepressor complex component SAP130 while assuming they investigated Mincle ligand SF3B3.

In total, we reviewed 70 articles mentioning either SF3B3, SAP130, their full names, or spliceosome-associated protein 130 (Figure 2). Out of 59 articles describing SF3B3, 34 papers referred to it as SAP130 (58%). The combination of a mistake in OMIM with a reference to a high-impact journal publication, and confusing product catalogues, has progressively led to mistakes in the literature. We may add that the lack of transparency in the materials and methods section amplified the issue addressed here. In total, 16 out of 70 papers reviewed (23%) did not report the materials in a manner that allowed verification. To illustrate, antibodies were often reported as ‘polyclonal rabbit anti-SAP130 (Abcam),’ without the accession number. Therefore, we could not check which antibody was used, whether it was raised against the splicing factor SF3B3, whether the incorrect abbreviation was used, or if it was indeed acting against SAP130. Six articles used the name spliceosome-associated protein 130 and SAP130 when referring to SF3B3, and the source of materials regarding the protein could not be retraced. Moreover, five articles had incomplete methods sections, and described SF3B3 as a histone deacetylase or Sin3A associated protein 130. Evidently, the authors were unaware of the distinction between the two proteins, and it could very well be that the researchers purchased antibodies and/or qPCR primers against SAP130 instead of the intended SF3B3. We contacted the corresponding authors of all 11 papers and requested the missing information regarding the antibodies, recombinant proteins, and/or qPCR primers (Table 1). Six authors responded with the requested accession numbers and/or qPCR primer sequences; these were correct for SF3B3. It cannot be confirmed that the research described by the non-responding research groups was performed with reagents designed for SF3B3.

## 4. Discussion

In summary, we discovered that the use of the incorrect gene symbol and abbreviation, spliceosome-associated protein 130 and SAP130, for splicing factor and Mincle ligand SF3B3, is persistent in the literature. This ambiguity has led to several mistakes in research on SF3B3. The protein was erroneously referred to as Sin3A associated protein 130 (SAP130), its function was described as histone deacetylase, and ultimately qPCR primers and an antibody against SAP130 were used in research on SF3B3 [1,2,3]. Furthermore, we cannot confirm that reagents used in five research articles were indeed for SF3B3, as the corresponding authors did not provide the missing reagent information [12,17,20,21,22]. The results in the former three papers should be validated, and the materials in the latter five papers should be checked to make sure they were designed for SF3B3 and not for SAP130. Moreover, databases, such as OMIM, and reagent suppliers need to correct their information about SF3B3 and SAP130.

This ambiguity of SF3B3 and SAP130 will persist if it is not called out and corrected. When possible, articles published online that can confirm the use of the right materials, should be edited to remove the description of SF3B3 as a histone deacetylase and perhaps use the correct abbreviation. Nature has informed us that they will publish a correction of the article by Seifert et al. [5], in which the reagent information will be added and the incorrect description of SF3B3 will be changed. We hope other journals will follow. In addition, all future research regarding SF3B3 should refrain from the alias spliceosome-associated protein 130 and its abbreviation SAP130. We have contacted Abcam to enquire about their antibodies against SF3B3 and SAP130 that were both called SAP130, and showed the information about, and UniProt accession numbers of, both proteins. They have improved the datasheets of these antibodies and confirmed the target of the antibodies we used here. We will contact OMIM and the other suppliers mentioned, in the hope that they will make a clear distinction between the proteins and refrain from using spliceosome-associated protein 130 and its abbreviation SAP130.

The frequent use of a gene synonym that is also the primary gene name of another gene, instead of the official gene symbol, is probably not limited to the use of SAP130 as an alias for SF3B3. In a data extraction from Ensembl BioMart of human gene symbols and gene synonyms as provided by HGNC, we found 437 genes with gene synonyms that are also the official gene symbol of another gene. When searching the literature for one of these genes using its gene synonym, it is possible to retrieve articles on another subject. Using such an article is a mistake that can be made, as evidenced by the confusion of SAP130 and SF3B3. Thus, careful attention to the use of official gene nomenclature is clearly warranted, as well as scrutinizing the information retrieved from the literature, databases, and companies.

## Figures and Tables

**Figure 1 cells-10-00590-f001:**
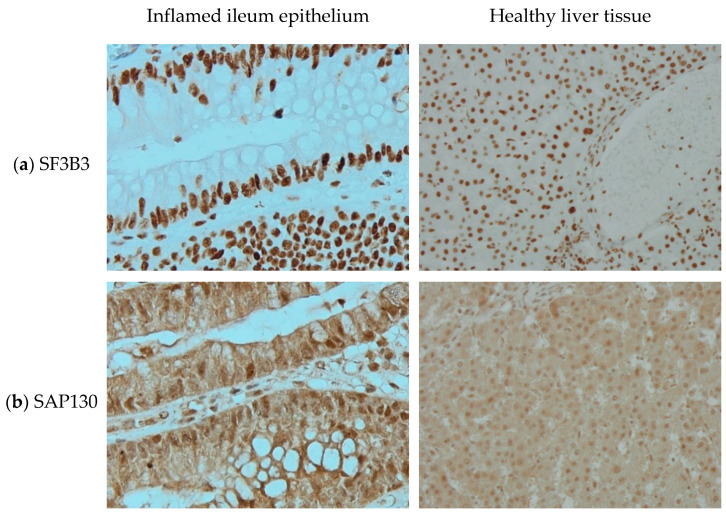
Immunohistochemistry staining of (**a**) splicing factor 3b subunit 3 (SF3B3) and (**b**) Sin3A associated protein 130 (SAP130) in (left) inflamed ileum epithelium, imaged at 40× magnification and (right) healthy liver tissue, imaged at 20× magnification.

**Figure 2 cells-10-00590-f002:**
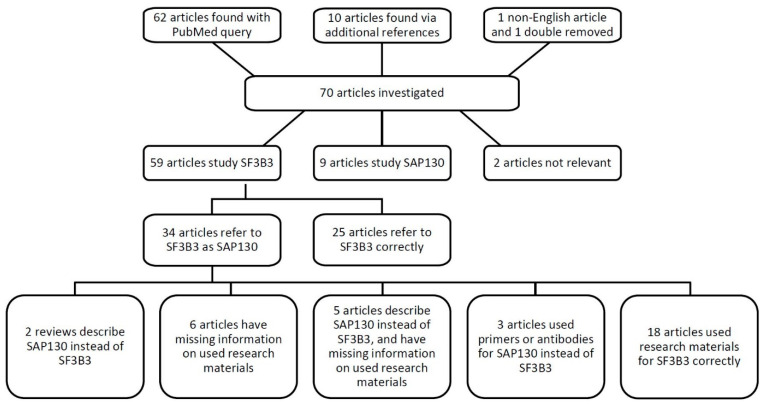
Schematic overview of the literature review of SF3B3 and SAP130.

**Table 1 cells-10-00590-t001:** List of papers whose corresponding authors were contacted. We asked eleven authors to provide missing antibody and recombinant protein ascension numbers and/or primer sequences. Three corresponding authors were notified of the mistakes in their papers.

Article	Journal	Name Used for SF3B3	Missing Information	Response
Suzuki 2013 [12]	Sci Rep	SAP130, a subunit of histone deacetylase	Antibody number	No response
Cordero-Espinoza 2013 [14]	Biol Open	Spliceosome-associated protein 130	SF3B3 protein number	Replied with correct number for SF3B3
de Rivero Vaccari 2015 [15]	J Neurotrauma	Sin3A associated protein 130, spliceosome-associated protein	2 antibody numbers, SF3B3 protein number	Replied with correct numbers for SF3B3
He 2015 [16]	Stroke	SAP130 (undefined)	Antibody and SF3B3 protein numbers	Replied with correct numbers for SF3B3
Seifert 2016 [5]	Nature	SAP130, a subunit of the histone deacetylase complex	4 antibodies and SF3B3 protein numbers, primer sequences	Replied with correct numbers and sequences for SF3B3. Nature is going to publish a correction
Greco 2016 [17]	J Immunol	Spliceosome-associated protein 130	3 antibody numbers	No response
Zhou 2016 [18]	Hepatology	Spliceosome-associated Protein 130	Antibody and SF3B3 protein numbers	Replied with correct numbers for SF3B3
Xie 2017 [19]	Brain Behav Immun	SAP130, a subunit of the histone deacetylase	2 antibody numbers, SF3B3 protein number	No response
Kim 2018 [20]	Am J Pathol	Spliceosome-associated protein 130	Antibody and SF3B3 protein numbers	No response
Wang 2019 [21]	J Cell Mol Med	Sin3A associated protein 130 kDa, subunit of the histone deacetylase	Antibody and SF3B3 protein numbers	No response
N’Diaye 2020 [22]	J Clin Invest	SAP130 (undefined)	Antibody number	Replied with correct number for SF3B3
Gong 2020 [1]	Ann NY Acad Sci	Spliceosome-associated protein 130	qPCR primers for SAP130	No response
Gong 2020 [2]	J Crohns Colitis	Spliceosome-associated protein 130	qPCR primers for SAP130	No response
Liu 2020 [3]	Ann Transl Med	SAP130, a subunit of histone deacetylase	Antibody against SAP130	No response

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
