# Peer review of "Ambiguity about Splicing Factor 3b Subunit 3 (SF3B3) and Sin3A Associated Protein 130 (SAP130)"

_cells, 2021, doi:10.3390/cells10030590_

Round 1
Reviewer 1 Report
In the manuscript by Paula I. Metselaar at., all, the authors performed a broad literature review of two apparently different proteins described as the same. It looks that careful check of the used by other authors and companies gene and protein names, abbreviations, references led the authors to the clear conclusion the SF3B3 and SAP130 are two distinct proteins and should be referred to in the existing and future literature according to the official nomenclature only. With the aid of the immunohistochemistry staining the author confirmed also the different localization of the proteins, SF3B3 is visible only in the nucleus, while SAP130 seems to be present more ubiquitously.
It is wort nothing that the authors already contacted the interested parties and convinced them to change the description.
Author Response
Response to Reviewer 1 Comments
Point 1: In the manuscript by Paula I. Metselaar at., all, the authors performed a broad literature review of two apparently different proteins described as the same. It looks that careful check of the used by other authors and companies gene and protein names, abbreviations, references led the authors to the clear conclusion the SF3B3 and SAP130 are two distinct proteins and should be referred to in the existing and future literature according to the official nomenclature only. With the aid of the immunohistochemistry staining the author confirmed also the different localization of the proteins, SF3B3 is visible only in the nucleus, while SAP130 seems to be present more ubiquitously.
It is wort nothing that the authors already contacted the interested parties and convinced them to change the description.

Response 1: We would like to thank the reviewer for the appreciation of our manuscript.
Reviewer 2 Report
The authors tried to explain the ambiguity of SF3B3 and Sin3A. They summerized the history of confusing literatures to make the names clear. They have done leterature review, contacting with selected corresponding authors. The only experiments they performed is the immunohistochemistry with antibodies to SF3B3 and SAP130. With this experiment, they demonstrate that these proteins are localized at different positions, so they are differents proteins. I do not thinks this manuscript is like a research article, with limited contents. Thus, it does not fit to Cells.
Author Response
Response to Reviewer 2 Comments
Point 1: The authors tried to explain the ambiguity of SF3B3 and Sin3A. They summerized the history of confusing literatures to make the names clear. They have done leterature review, contacting with selected corresponding authors. The only experiments they performed is the immunohistochemistry with antibodies to SF3B3 and SAP130. With this experiment, they demonstrate that these proteins are localized at different positions, so they are differents proteins. I do not thinks this manuscript is like a research article, with limited contents. Thus, it does not fit to Cells.
Response 1: We thank the reviewer for expressing his/her views. While we agree with the notion that our manuscript does not provide much experimental data, we respectfully disagree that our manuscript does not fit to Cells as we believe it is relevant to its readership. In acknowledgement of the critical point raised by the reviewer, we resubmitted the manuscript as a Communication instead of an Article.
Revisions in the manuscript: Line 1, change of Article Type from Article to Communication.
Reviewer 3 Report
The paper describes a confusing observation that one of aliases of gene SF3B3 (splicing factor 3b subunit 3) is SAP130 that in turn is a primary name of completely different gene - Sin3A associated protein 130. Authors shown that this ambiguity propagated in scientific databases and catalogues of biochem companies and have already caused researchers to mistakenly use primers or antibodies for SAP130 to investigate SF3B3. Authors attempted to contact everybody involved in this discrepancy including paper authors and database owners. I believe that this work is significant effort toward less chaotic biology.
While in general I’m satisfied by paper quality I have two suggestions
1. To my mind abstract is not very clear when it describes the problem. Particularly, first it says that “protein investigated, however, was splicing factor 3b subunit 3 (SF3B3)” but than “ while in fact SAP130 was investigated”. I assume that mentioned researchers planned to investigate SF3B3 but actually used primers and/or antibodies for SAP130, right? If so, I think that abstract should be reformulated to make it absolutely clear.
2. I think it would be great to put the research into broader context. My brief analysis of human gene names and synonyms downloaded from Ensembl BioMart showed that there are about 500 gene pairs where one gene synonym is primary gene name of another. Probably that problem similar to the one described in the paper also take place with (for example) CYP11B1 (cytochrome P450 family 11 subfamily B member 1; also known as CPN1) and CPN1 (carboxypeptidase N subunit 1), on a web page (https://www.biocompare.com/9776-Antibodies/7400254-CPN1-Carboxypeptidase-N-Catalytic-Chain-CPN-Anaphylatoxin-inactivator-Arginine-carboxypeptidase-Carboxypeptidase-N-Polypeptide-1-Carboxypeptidase-N-Small-Subunit-Kininase-1-Lysine-carboxypeptidase-Plasma-carboxypeptidase-B-Serum/?pda=9776|7400254_0_0||3|221858) CYP11B1 is mentioned among a targets of Rabbit Anti-CPN1 antibodies. So, I would suggest to at least discuss possible presence of other such cases
Author Response
Response to Reviewer 3 Comments
The paper describes a confusing observation that one of aliases of gene SF3B3 (splicing factor 3b subunit 3) is SAP130 that in turn is a primary name of completely different gene - Sin3A associated protein 130. Authors shown that this ambiguity propagated in scientific databases and catalogues of biochem companies and have already caused researchers to mistakenly use primers or antibodies for SAP130 to investigate SF3B3. Authors attempted to contact everybody involved in this discrepancy including paper authors and database owners. I believe that this work is significant effort toward less chaotic biology.
While in general I’m satisfied by paper quality I have two suggestions
Point 1: To my mind abstract is not very clear when it describes the problem. Particularly, first it says that “protein investigated, however, was splicing factor 3b subunit 3 (SF3B3)” but than “ while in fact SAP130 was investigated”. I assume that mentioned researchers planned to investigate SF3B3 but actually used primers and/or antibodies for SAP130, right? If so, I think that abstract should be reformulated to make it absolutely clear.
Response 1: We thank the reviewer for the appreciation of our manuscript and the helpful comments. We acknowledge the unclear description of the two proteins in the abstract and we have reformulated the sentences to provide a better explanation of the problem.
Revisions in the manuscript: Lines 13-19, change of first sentences of Abstract from the prior version:
In 2020, three articles were published on a protein that can activate the immune system by binding to Macrophage-inducible C-type lectin receptor (Mincle), referred to as 'SAP130, a subunit of the histone deacetylase complex.' The protein investigated, however, was splicing factor 3b subunit 3 (SF3B3). This splicing factor is unrelated to SAP130 (Sin3A associated protein 130; a subunit of the histone deacetylase-dependent Sin3A corepressor complex). The conclusions in these articles were formulated for SF3B3, while in fact SAP130 was investigated with qPCR primers designed for the encoding gene, and antibodies against the protein.
To the adapted version:
In 2020, three articles were published on a protein that can activate the immune system by binding to Macrophage-inducible C-type lectin receptor (Mincle). In the articles, the protein was referred to as 'SAP130, a subunit of the histone deacetylase complex.' However, the Mincle ligand the authors aimed to investigate is splicing factor 3b subunit 3 (SF3B3). This splicing factor is unrelated to SAP130 (Sin3A associated protein 130; a subunit of the histone deacetylase-dependent Sin3A corepressor complex). The conclusions in the three articles were formulated for SF3B3, while the researchers used qPCR primers and antibodies against SAP130.
Point 2: I think it would be great to put the research into broader context. My brief analysis of human gene names and synonyms downloaded from Ensembl BioMart showed that there are about 500 gene pairs where one gene synonym is primary gene name of another. Probably that problem similar to the one described in the paper also take place with (for example) CYP11B1 (cytochrome P450 family 11 subfamily B member 1; also known as CPN1) and CPN1 (carboxypeptidase N subunit 1), on a web page (https://www.biocompare.com/9776-Antibodies/7400254-CPN1-Carboxypeptidase-N-Catalytic-Chain-CPN-Anaphylatoxin-inactivator-Arginine-carboxypeptidase-Carboxypeptidase-N-Polypeptide-1-Carboxypeptidase-N-Small-Subunit-Kininase-1-Lysine-carboxypeptidase-Plasma-carboxypeptidase-B-Serum/?pda=9776|7400254_0_0||3|221858) CYP11B1 is mentioned among a targets of Rabbit Anti-CPN1 antibodies. So, I would suggest to at least discuss possible presence of other such cases.
Response 2: We would like to thank the reviewer for the suggestion to put our discovery in a broader context. We have repeated the reviewer’s analysis of gene names and synonyms (HGNC based) downloaded from BioMart, and indeed found 437 gene pairs. We have added a paragraph to the discussion in which we discuss the possible presence of other cases such as we described.
Revisions in the manuscript: Lines 228-237, addition of the following paragraph on potential other cases:
The frequent use of a gene synonym that is also the primary gene name of another gene, instead of the official gene symbol, is probably not limited to the use of SAP130 as an alias for SF3B3. In a data extraction from Ensembl BioMart of human gene symbols and gene synonyms as provided by HGNC, we found 437 genes with gene synonyms that are also the official gene symbol of another gene. When searching the literature for one of these genes using its gene synonym, it is possible to retrieve articles on another subject. Using such an article is a mistake that can be made, as evidenced by the confusion of SAP130 and SF3B3. Thus, careful attention to the use of official gene nomenclature is clearly warranted, as well as scrutinizing the information retrieved from the literature, databases, and companies.
Round 2
Reviewer 2 Report
The manuscript did not make significant progress in the revision stage. The authors changed the format from "article" to "communication". However, I feel that "communication" should be more important findings. The manuscript still does not fit to "cells".